# Inflammation and Oxidative Stress Gene Variability in Retinal Detachment Patients with and without Proliferative Vitreoretinopathy

**DOI:** 10.3390/genes14040804

**Published:** 2023-03-27

**Authors:** Xhevat Lumi, Filippo Confalonieri, Metka Ravnik-Glavač, Katja Goričar, Tanja Blagus, Vita Dolžan, Goran Petrovski, Marko Hawlina, Damjan Glavač

**Affiliations:** 1Eye Hospital, University Medical Centre Ljubljana, 1000 Ljubljana, Slovenia; xhevat.lumi@kclj.si (X.L.); marko.hawlina@kclj.si (M.H.); 2Center for Eye Research and Innovative Diagnostics, Department of Ophthalmology, Oslo University Hospital, 0450 Oslo, Norway; filippo.confalonieri01@gmail.com (F.C.); goran.petrovski@medisin.uio.no (G.P.); 3Institute for Clinical Medicine, Faculty of Medicine, University of Oslo, 0315 Oslo, Norway; 4Department of Biomedical Sciences, Humanitas University, 20090 Milan, Italy; 5Institute of Biochemistry and Molecular Genetics, Faculty of Medicine, University of Ljubljana, 1000 Ljubljana, Slovenia; metka.ravnik-glavac@mf.uni-lj.si (M.R.-G.); katja.goricar@mf.uni-lj.si (K.G.); tanja.blagus@mf.uni-lj.si (T.B.); vita.dolzan@mf.uni-lj.si (V.D.); 6Department of Ophthalmology, School of Medicine, University of Split, University Hospital Centre, 21 000 Split, Croatia; 7Department of Molecular Genetics, Institute of Pathology, Faculty of Medicine, University of Ljubljana, 1000 Ljubljana, Slovenia; 8Center for Human Genetics & Pharmacogenomics, Faculty of Medicine, University of Maribor, 2000 Maribor, Slovenia

**Keywords:** rhegmatogenous retinal detachment, proliferative vitreoretinopathy, single nucleotide polymorphism, SNP, genotyping, association study, oxidative stress

## Abstract

This study investigated the association between certain genetic variations and the risk of developing proliferative vitreoretinopathy (PVR) after surgery. The study was conducted on 192 patients with primary rhegmatogenous retinal detachment (RRD) who underwent 3-port pars plana vitrectomy (PPV). The distribution of single nucleotide polymorphisms (SNPs) located in genes involved in inflammation and oxidative stress associated with PVR pathways were analyzed among patients with and without postoperative PVR grade C1 or higher. A total of 7 defined SNPs of 5 genes were selected for genotyping: rs4880 (*SOD2*); rs1001179 (*CAT*); rs1050450 (*GPX1*); rs1143623, rs16944, rs1071676 (*IL1B*); rs2910164 (*MIR146A*) using competitive allele-specific polymerase chain reaction. The association of SNPs with PVR risk was evaluated using logistic regression. Furthermore, the possible association of SNPs with postoperative clinical parameters was evaluated using non-parametric tests. The difference between two genotype frequencies between patients with or without PVR grade C1 or higher was found to be statistically significant: *SOD2* rs4880 and *IL1B* rs1071676. Carriers of at least one polymorphic *IL1B* rs1071676 GG allele appeared to have better postoperative best-corrected visual acuity only in patients without PVR (*p* = 0.070). Our study suggests that certain genetic variations may play a role in the development of PVR after surgery. These findings may have important implications for identifying patients at higher risk for PVR and developing new treatments.

## 1. Introduction

Proliferative vitreoretinopathy (PVR) represents the proliferation and contraction of fibrocellular membranes over and under the neurosensory retina and within the vitreous cavity in patients with rhegmatogenous retinal detachment (RRD) [1,2]. It is the main complication following retinal detachment (RD) surgery and the leading cause of failure in the management of RRD [3,4,5,6,7]. It is estimated that PVR occurs in 5–10% of patients with RRD [8,9,10,11]. According to the Retina Society Terminology Committee’s classification, the most severe stage of PVR is grade C, which is characterized by starfolds and contraction of the retina [6].

The processes leading to PVR development are complex. PVR is driven by a wide spectrum of growth factors and cytokines [5,7]. Numerous molecules related to the inflammatory process have been implicated in the PVR’s development, such as growth factors, transforming growth factors, interleukins, chemokines, and tumor necrosis factor [11,12,13,14,15,16,17]. Among the many inflammatory mediators, transforming growth factor β (TGF-β), platelet-derived growth factor (PDGF), interleukin (IL)-6 (IL-6), IL-8, and tumor necrosis factor α (TNF-α) are thought to be particularly relevant. Other growth factors and cytokines, including basic fibroblast growth factor (bFGF), vascular endothelial growth factor (VEGF), IL-1 α, IL-2, IL-3, and intercellular adhesion molecule-1 (ICAM-1) have also been found to play an important role in the PVR processes [7,14,15,16,17,18,19].

As in many other diseases, in RD and during the process of PVR formation, oxidative stress and inflammation are important factors that lead to disturbed redox signaling and molecular damage [20]. Higher levels of reactive oxygen species (ROS), which affect many physiological processes, have been described in retinal diseases, including PVR [21]. Oxidative stress in the cellular environment can also lead to dysregulated para-inflammation, which further gives rise to inflammatory processes mediated by pro-inflammatory cytokines such as TNF-α, IL-6, and IL-1β [21]. In studies on the neuroretina in mice, it has been shown that the increase of para-inflammatory responses is characterized by a further increase of ROS production in the retinal pigment epithelium (RPE), microglial activation and subretinal migration, as well as breakdown of the blood-retinal barrier [22,23]. These changes have also been described as occurring during PVR formation [24]. Given the important involvement of oxidative stress in many pathogenic pathways, it has been postulated that antioxidant activity may reduce photoreceptor loss following RD and delay development to PVR [21,25,26,27].

To expand the current knowledge on the association of selected single nucleotide polymorphisms (SNPs) with PVR, we investigated seven selected SNPs located within five genes (*SOD2*, *CAT*, *GPX1*, *IL1B*, and *MIR146A*) likely involved in the PVR pathways (oxidative stress and inflammation) in RRD patients with and without postoperative PVR grade C1 or higher.

## 2. Materials and Methods

### 2.1. Study Population

A genetic association study was conducted on 192 patients with primary RRD who underwent 3-port pars plana vitrectomy (PPV) at the Eye Hospital, University Medical Centre Ljubljana, Slovenia. The study was approved by the National Medical Ethics Committee of the Republic of Slovenia (Approval No. 91/05/11) and conducted in accordance with the Declaration of Helsinki. All patients provided written informed consent and agreed to the genetic analysis of SNPs potentially associated with the PVR formation process.

Only patients after PPV and laser retinopexy for RRD were included in the study. Exclusion criteria were as follows: tractional or posttraumatic RD, RD secondary to a macular hole, cryotherapy performed during PPV, PPV combined with a scleral buckle, diabetic retinopathy, central- or branch-retinal vein occlusion, glaucoma, and age under 16 years.

The PVR stage was graded according to the updated classification of the Retina Society Terminology Committee (1991) [6]. Patients who developed PVR Grade C1 or higher within 3 months after the primary RRD surgery were enrolled as PVR cases, and patients who did not develop PVR Grade C1 or higher were enrolled as controls (or non-PVR cases).

Additional data collected from the patient’s records were: age at the time of surgery, gender, duration of symptoms, Snellen best corrected visual acuity (BCVA), macular status (on/off), intraocular pressure (IOP), and central retinal thickness (CRT). A Topcon swept-source optical coherence tomography (SS-OCT) was used to obtain the OCT images (SS-OCT, DRI OCT Triton, Topcon, Tokyo, Japan). The horizontal cross-sectional B-scan image was recorded at the postoperative visit 6 months after the surgery. CRT was measured using the software-based measurement tool.

### 2.2. Definition of Clinical Parameters

To evaluate the association of selected SNPs with clinical phenotypes, the relevant clinical parameters had to be identified first. These were defined as postoperative BCVA, IOP, and CRT. For each of these parameters, the association with SNPs was evaluated in the groups without and with PVR grade C1 separately.

### 2.3. Blood Collection and DNA Extraction

DNA was extracted from six milliliters of peripheral blood using the QIAamp DNA Blood Midi Kit (QIAGEN, Hilden, Germany), according to the manufacturer’s instructions. Extracted DNA was stored at −20 °C until used for genotyping.

### 2.4. Genotyping of PVR Patients

Five candidate genes were selected based on their direct involvement in response to oxidative stress and inflammation pathways. Only common, experimentally confirmed or in silico predicted functional polymorphisms with minor allele frequency above 5% in the European population were included in the study: non-synonymous SNPs located within the coding region or SNPs in the vicinity of selected genes (5′ and 3′ untranslated regions) that could alter expression levels. Seven SNPs fulfilling these criteria were selected for genotyping: rs4880 (*SOD2*); rs1001179 (*CAT*); rs1050450 (*GPX1*); rs1143623, rs16944, rs1071676 (*IL1B*); and rs2910164 (*MIR146A*). Genotypes were determined using competitive allele-specific polymerase chain reaction (KASP, LGC Genomics, Hoddesdon, UK) according to the manufacturer’s instructions.

### 2.5. Statistical Analysis

Categorical variables were described using frequencies, while continuous variables were described with a median and a 25–75% range. Fisher’s exact test and the Mann–Whitney test were used to compare the distribution of categorical and continuous variables between PVR cases and controls, respectively. For all selected SNPs, deviation from the Hardy–Weinberg equilibrium (HWE) was evaluated using the chi-square test. Both dominant and additive genetic models were used in the analysis. To evaluate the association of SNPs with PVR risk, odds ratios (ORs) with 95% confidence intervals (95% CIs) were calculated using logistic regression. The Mann–Whitney or Kruskal–Wallis tests were used to evaluate the association of SNPs with continuous clinical parameters. All tests were two-sided. As 7 SNPs were included in the analyses, the Bonferroni correction was used to account for multiple comparisons: *p*-values below 0.007 were considered statistically significant, while *p*-values between 0.006 and 0.050 were considered nominally significant. All analyses were performed using IBM SPSS Statistics version 27.0 (IBM Corporation, Armonk, NY, USA).

## 3. Results

A total of 192 patients with new-onset RD in one eye were enrolled, of whom 116 (60.4%) were males and 76 (39.6%) were females. Macula-off detachments represented 138 (72.3%) and macula-on represented 53 (27.7%) of the cases. Included in total were 112 (58.3%) patients who developed PVR Grade C1 or higher within 3 months after primary surgery for RRD and 80 (41.7%) patients who did not develop PVR. The clinical characteristics of the whole study group and comparison between patients with or without PVR grade C1 or higher are presented in Table 1. In the non-PVR group, there were more males compared to the PVR group, but the difference was not statistically significant (*p* = 0.052). Patients in the PVR group were also slightly older (65 (55–74) vs. 60 (51–70.8) years) in the non-PVR group (*p* = 0.081)). The distribution of patients with RRD in the left or right eye was similar between both groups (*p* = 0.656).

The number of surgeries in the whole cohort ranged from 1 to 5. Most of the patients in the non-PVR group (97.5%) had only one surgery, with only two patients in this group having two surgeries, whereas most patients with PVR grade C1 or higher (71.4%) received more than one surgery (Table 1). The median duration of symptoms in the whole cohort was 14 (7–30) days. In the non-PVR group, the median duration of symptoms was 8 (5–14) days, while it was significantly longer in the PVR group (14 (7–30)) days, *p* < 0.001).

Preoperative median BCVA in the non-PVR group was 0.1 (0.01–0.5), whereas in patients with PVR grade C1 or higher it was 0.01 (range 0.01–0.2), which was significantly lower (*p* < 0.001). The same statistically significant difference was maintained postoperatively (0.7 vs. 0.2, respectively, for the non-PVR versus PVR groups, *p* < 0.001) (Table 1). Patients with PVR grade C1 or higher also had significantly higher CRT measured by OCT (281 µm vs. 245 µm, *p* = 0.008). Preoperative IOP was slightly lower in patients with PVR grade C1 or higher (*p* = 0.028), but no differences could be observed postoperatively (Table 1).

Genotype frequencies of 7 selected SNPs located within or in the vicinity of 5 genes involved in the oxidative stress (*SOD2*, *CAT*, *GPX1*) and inflammatory (*IL1B*, *MIR146A*) pathways among all patients with RRD are presented in Table 2. The genotype frequencies for all SNPs were in agreement with HWE.

Comparing the genotype frequencies between patients with or without PVR grade C1 or higher, only 2 SNPs were found to be nominally significant: *SOD2* rs4880 and *IL1B* rs1071676 (Table 3). Carriers of the two polymorphic *SOD2* rs4880 alleles (TT genotype) were more likely to have higher grade PVR (OR = 2.31, 95% CI = 1.01–5.27, *p* = 0.046) compared to the CC genotype, even after adjustment for age and gender (OR = 2.74, 95% CI = 1.17–6.42, *p* = 0.021). In the dominant model, carriers of at least one polymorphic *SOD2* rs4880 T allele appeared to have higher-grade PVR only after adjustment for age and gender (OR = 1.97, 95% CI = 1.00–3.90, *p* = 0.050). On the other hand, carriers of the single polymorphic *IL1B* rs1071676 allele (GC genotype) were less likely to have higher grade PVR only after adjustment for age and gender (OR = 0.50, 95% CI = 0.26–0.95, *p* = 0.034), while no differences could be observed in the univariable analysis or in the dominant model (all *p* > 0.050) (Table 3). None of the associations reached the threshold for statistical significance after adjustment for multiple comparisons.

The association of selected SNPs with postoperative clinical parameters among the RRD patients with and without PVR grade C1 or higher is shown in Table 4. None of the investigated SNPs were significantly associated with any of the clinical parameters in both groups. However, carriers of at least one polymorphic *IL1B* rs1071676 GG allele appeared to have better postoperative BCVA only in patients without PVR (*p* = 0.070), although none of the associations were statistically significant after adjustment for multiple comparisons.

## 4. Discussion

Several investigations have tried to elucidate the question of whether genetic factors play a key role in the wound healing process and the development of PVR [28]. Some of them have identified specific SNPs located in genes involved in the PVR pathways, which may be involved in the disease pathogenesis and represent potential predictive factors for PVR development [9,29,30,31,32,33,34,35]. Some genetic association studies performed in the last two decades have also highlighted the potential of various gene polymorphisms of inflammatory mediators and growth factors as new biomarkers in PVR pathogenesis as well as a potential tool in its diagnosis and treatment [9,29,30,32,36,37].

In a previous study, we found differences in genotype distributions between patients with and without PVR at rs1800795 of the *IL6* gene, as well as at rs1800871 in the vicinity of the *IL10* gene and rs1800471 in the *TGFB1* gene [35].

The *SOD2* rs4880 T allele and *IL1B* rs1071676 GC genotypes were nominally associated with increased risk for developing PVR grade C1 or higher, but the associations could not reach statistical significance after adjustment for multiple comparisons. The SOD2 enzyme can be considered to have a potential role in the development of PVR by influencing the oxidant state of the vitreous body [38,39]. The non-synonymous SNP rs4880 has been linked to poor health outcomes and increased risk for breast cancer, cardiovascular disease, and diabetes mellitus [40,41,42], but it has so far not yet been investigated in PVR. The *IL1B* gene, located on chromosome 2, encodes *IL-1β*, an important mediator of the inflammatory response, which is involved in a variety of cellular processes, including cell proliferation, differentiation, and apoptosis [43], making it a pivotal mediator for PVR development [44].

Our findings regarding these two SNPs may be indicative of their role in PVR, as both *SOD2* and *IL1B* genes are involved in the inflammatory processes present in PVR, as well as regulation of the inflammatory response or oxidative stress, but further studies are needed to confirm our results at a larger scale. When the groups of patients were compared against their clinical parameters, statistically significant differences were found for the pre- and post-operative BCVA, macula on/off status, the number of surgeries performed, the duration of symptoms, the post-operative CRT, and the pre-operative IOP. However, no associations were observed between the investigated SNPs and the postoperative clinical parameters collected.

In a previous study, we also found a statistically significant difference in the genotype distributions between healthy controls and PVR patients for the rs1800629 in *TNF* (c.-308G>A) [35]. Previously, the TNF locus, which encodes also for *TNF-α*, has been investigated in three subsequent studies by a group in Spain and shown to be associated with increased risk of PVR development, including that of rs1800629 [8,9,29]. Previous investigations of the genotype distribution in the Slovenian population have shown a statistically significant difference between patients with and without PVR and a potential association between PVR and SNPs located within the *IL-6* and *TGFB1* genes, and in the vicinity of the gene for *IL-10*. However, after adjustment for multiple hypothesis testing, none of the comparisons appeared to be statistically significant [34]. Our previous analysis has shown a wide variation in the prevalence of genotypes among different ethnic groups, with a great limitation being the small groups of patients analyzed [34,35].

Altogether, more than 40 SNPs have been previously associated with PVR [35], but only 17 of them in 12 genes have remained associated with PVR after adjustment for multiple comparisons [14,19,20,21,22,23]. Moreover, several associations between SNP genotypes and the PVR phenotype could not be replicated in studies conducted on groups of different ethnic backgrounds within European populations [34,35]. The likelihood for the detection of false positive results when those are not adjusted properly is high and represents the most obvious problem in gene-disease association studies [45,46,47,48]. To confirm the association between SNPs and a disease, replication among different study populations is crucial, as differences in genotype frequencies might contribute to the results [49].

In the present study, a genetic association and analysis of correlation to clinical parameters was conducted on 192 patients with primary RRD who underwent PPV and revealed 2 genotypic associations to the PVR group in *SOD2* rs4880 and *IL1B* rs1071676 SNPs. When comparing the groups in terms of clinical parameters, we found significant differences between them in most of the analyzed parameters. Our results demonstrate an association between the development of PVR C1 or higher in the *SOD2* rs4880 TT genotype compared to the CC genotype, as well as the *IL1B* rs1071676 GC genotype compared to the GG genotype. To our knowledge, this finding is the first in the analysis of PVR-related gene polymorphisms that links PVR to *SOD2* and *IL1B* gene polymorphisms. Our results suggest that reducing the level of oxidative stress and influencing the cascade of inflammatory responses within the vitreous cavity can possibly change the course of PVR formation. If we further speculate, by manipulating the vitreous antioxidant profile, one could protect the eye from oxidative stress, which is responsible for the exacerbation of the inflammatory response that leads to PVR development [50]. Within the speculation limits, both SNPs may serve as new biomarkers for an unfavorable course of RRD treatment.

This study has some limitations. At this stage, we could not include a true control group of a healthy population. It could be possible to eliminate bias and inaccuracies from such research by comparing RRD patients with healthy controls and expanding the investigation to a larger sample size.

## 5. Conclusions

We hereby analyzed the potential association of SNPs of certain genes associated with PVR formation. A statistically significant difference between patients with and without PVR was found in SNPs of the *SOD2* and *IL1B* genes. The aim in PVR research, as well as in other human diseases, is to detect genetic associations, which can be replicated in future studies without a significant bias. Our study confirmed that differences in the genotype distributions exist. Due to the limited number of patients enrolled, the present study is based only on a single center-Slovenian population. It is crucial to conduct larger multicenter and possibly population-based studies in the near future to further strengthen our findings and genetic association results.

## Figures and Tables

**Table 1 genes-14-00804-t001:** Clinical characteristics of the patients with RRD and comparison of the patients with or without PVR.

Characteristic		Total Number of Patients (N = 192)	Patients without PVR (N = 80)	Patients with PVR Grade C (N = 112)	*p*
Gender	Male, N (%)	116 (60.4)	55 (68.8)	61 (54.5)	0.052 ^a^
	Female, N (%)	76 (39.6)	25 (31.3)	51 (45.5)	
Age	Years, Median (25–75%)	64 (53–72)	60 (51–70.8)	65 (55–74)	0.081 ^b^
Eye	Right, N (%)	109 (57.1) [1]	47 (59.5) [1]	62 (55.4)	0.656 ^a^
	Left, N (%)	82 (42.9)	32 (40.5)	50 (44.6)	
PVR grade	A, N (%)	32 (16.7)	32 (40.0)	0 (0.0)	<0.001 ^a^
	B, N (%)	48 (25.0)	48 (60.0)	0 (0.0)	
	C, N (%)	112 (58.3)	0 (0.0)	112 (100.0)	
Preoperative BCVA	Median (25–75%)	0.01 (0.01–0.30) [5]	0.10 (0.01–0.50) [4]	0.01 (0.01–0.20) [1]	<0.001 ^b^
Postoperative BCVA	Median (25–75%)	0.30 (0.10–0.70) [1]	0.70 (0.50–1.00) [1]	0.20 (0.01–0.40)	<0.001 ^b^
Macula	Off, N (%)	138 (72.3) [1]	50 (63.3) [1]	88 (78.6)	0.023 ^a^
	On, N (%)	53 (27.7)	29 (36.7)	24 (21.4)	
Number of surgeries	1, N (%)	109 (57.1) [1]	77 (97.5) [1]	32 (28.6)	<0.001 ^a^
	2, N (%)	42 (22.0)	2 (2.5)	40 (35.7)	
	3, N (%)	23 (12.0)	0 (0.0)	23 (20.5)	
	4, N (%)	14 (7.3)	0 (0.0)	14 (12.5)	
	5, N (%)	3 (1.6)	0 (0.0)	3 (2.7)	
Duration of symptoms	Days, Median (25–75%)	14 (7–30) [2]	8 (5–14) [2]	14 (7–30)	<0.001 ^b^
OCT-CRT	µm, Median (25–75%)	262.5 (223.3–321.5) [16]	245 (218–286) [5]	281 (230.5–389) [11]	0.008 ^b^
Preoperative IOP	mmHg, Median (25–75%)	13 (12–16) [23]	14 (12–17.3) [6]	13 (10–16) [17]	0.028 ^b^
Postoperative IOP	mmHg, Median (25–75%)	15 (12–17) [5]	15 (13–17) [1]	14.5 (11–18) [4]	0.426 ^b^

The number of missing data is presented in [ ] brackets. ^a^ calculated using Fisher’s exact test; ^b^ calculated using the Mann–Whitney test. BCVA—best corrected visual acuity; IOP—intraocular pressure; OCT-CRT—optical coherence tomography central retinal thickness; PVR—proliferative vitreoretinopathy; RRD—rhegmatogenous retinal detachment.

**Table 2 genes-14-00804-t002:** Genotype frequencies of selected polymorphisms among all patients with RRD.

Gene	Polymorphism	Role	Genotype	N (%)	VAF	pHWE	VAF (1000 G)	*p* *
*SOD2*	rs4880	p.Ala16Val	CC	46 (24.1) [1]	51.0	0.947	53.4	0.365
			CT	95 (49.7)				
			TT	50 (26.2)				
*CAT*	rs1001179	c.-262C>T	CC	115 (60.2) [1]	22.5	0.895	23.5	0.657
			CT	66 (34.6)				
			TT	10 (5.2)				
*GPX1*	rs1050450	p.Pro198Leu	CC	105 (54.7)	26.8	0.422	33.6	0.006
			CT	71 (37.0)				
			TT	16 (8.3)				
*IL1B*	rs1143623	c.-1560G>C	GG	102 (53.1)	28.0	0.499	28.7	0.770
			GC	74 (38.0)				
			CC	17 (8.9)				
	rs16944	c.-598T>C	TT	30 (15.6) [1]	64.1	0.087	65.0	0.722
			TC	77 (40.3)				
			CC	84 (44.0)				
	rs1071676	c.*505G>C	GG	110 (57.3)	26.0	0.062	24.9	0.631
			GC	64 (33.3)				
			CC	18 (9.4)				
*MIR146A*	rs2910164	n.60G>C	GG	122 (63.5)	20.3	0.972	23.5	0.153
			GC	62 (32.3)				
			CC	8 (4.2)				

The number of missing data is presented in [ ] brackets. HWE—Hardy–Weinberg equilibrium; VAF—variant allele frequency; RRD—rhegmatogenous retinal detachment. VAF (1000 G): variant allele frequency reported in European population in the 1000 Genomes project in dbSNP. * comparison of variant allele frequency with 1000 Genomes project using the chi-square test.

**Table 3 genes-14-00804-t003:** Comparison of genotype frequencies between patients with and without PVR grade C1 or higher at 3 months postoperatively.

SNP	Genotype	Cases without PVR Grade C1 N (%)	Cases with PVR Grade C1 N (%)	OR (95% CI)	*p*	OR (95% CI)_adj_	*p_adj_*
*SOD2* rs4880	CC	25 (54.3)	21 (45.7)	Reference			
	CT	38 (40)	57 (60)	1.79 (0.88–3.63)	0.110	1.67 (0.81–3.46)	0.165
	TT	17 (34)	33 (66)	2.31 (1.01–5.27)	0.046	2.74 (1.17–6.42)	0.021
	CT + TT	55 (37.9)	90 (62.1)	1.95 (1.00–3.81)	0.051	1.97 (1.00–3.90)	0.050
*CAT* rs1001179	CC	44 (38.3)	71 (61.7)	Reference			
	CT	30 (45.5)	36 (54.5)	0.74 (0.40–1.37)	0.344	0.74 (0.39–1.38)	0.337
	TT	5 (50)	5 (50)	0.62 (0.17–2.26)	0.469	0.66 (0.18–2.44)	0.531
	CT + TT	35 (46.1)	41 (53.9)	0.73 (0.40–1.31)	0.285	0.72 (0.40–1.32)	0.292
*GPX1* rs1050450	CC	45 (42.9)	60 (57.1)	Reference			
	CT	28 (39.4)	43 (60.6)	1.15 (0.62–2.13)	0.651	1.14 (0.61–2.13)	0.673
	TT	7 (43.8)	9 (56.3)	0.96 (0.33–2.79)	0.946	1.07 (0.36–3.15)	0.907
	CT + TT	35 (40.2)	52 (59.8)	1.11 (0.63–1.98)	0.713	1.13 (0.63–2.03)	0.685
*IL1B* rs1143623	GG	43 (42.2)	59 (57.8)	Reference			
	GC	30 (41.1)	43 (58.9)	1.04 (0.57–1.92)	0.888	1.09 (0.58–2.02)	0.792
	CC	7 (41.2)	10 (58.8)	1.04 (0.37–2.95)	0.940	1.03 (0.36–2.99)	0.950
	GC + CC	37 (41.1)	53 (58.9)	1.04 (0.59–1.86)	0.883	1.08 (0.60–1.93)	0.804
*IL1B* rs16944	TT	13 (43.3)	17 (56.7)	Reference			
	TC	36 (46.8)	41 (53.2)	0.87 (0.37–2.04)	0.750	0.86 (0.36–2.03)	0.729
	CC	30 (35.7)	54 (64.3)	1.38 (0.59–3.22)	0.461	1.34 (0.57–3.17)	0.502
	TC + CC	66 (41)	95 (59)	1.10(0.50–0.42 pm)	0.811	1.08 (0.49–2.40)	0.852
*IL1B* rs1071676	GG	42 (38.2)	68 (61.8)	Reference			
	GC	34 (53.1)	30 (46.9)	0.54 (0.29–1.02)	0.056	0.50 (0.26–0.95)	0.034
	CC	4 (22.2)	14 (77.8)	2.16 (0.67–7.01)	0.199	2.47 (0.75–8.19)	0.139
	GC + CC	38 (46.3)	44 (53.7)	0.72 (0.40–1.28)	0.257	0.69 (0.38–1.25)	0.224
*MIR146A* rs2910164	GG	56 (45.9)	66 (54.1)	Reference			
	GC	21 (33.9)	41 (66.1)	1.66 (0.88–3.13)	0.119	1.69 (0.89–3.23)	0.110
	CC	3 (37.5)	5 (62.5)	1.41 (0.32–6.18)	0.645	1.45 (0.33–6.41)	0.625
	GC + CC	24 (34.3)	46 (65.7)	1.63 (0.88–2.99)	0.117	1.66 (0.90–3.08)	0.107

Adj—adjusted for age and gender.

**Table 4 genes-14-00804-t004:** Association of selected polymorphisms with postoperative clinical parameters in RRD patients with and without PVR grade C1 or higher.

		Cases without PVR						Cases with PVR Grade C1 or Higher					
SNP	Genotype	BCVA Median (25–75%)	*p*	OCT-CRT Median (25–75%)	*p*	IOP Median (25–75%)	*p*	BCVA Median (25–75%)	*p*	OCT-CRT Median (25–75%)	*p*	IOP Median (25–75%)	*p*
*SOD2* rs4880	CC	0.7 (0.4–1)	0.940 ^a^	232.5 (200–263.5)	0.170 ^a^	15 (13–17)	0.943 ^a^	0.2 (0.04–0.4)	0.192 ^a^	304 (238.5–516)	0.176 ^a^	14 (9.5–17)	0.600 ^a^
	CT	0.7 (0.55–0.95)		262.5 (217.3–310.8)		15 (13.5–17)		0.2 (0.03–0.4)		267 (205.5–366)		14 (11–17)	
	TT	0.7 (0.4–1)		248 (233.5–286.5)		15 (12–18)		0.1 (0.01–0.2)		283 (233.5–463.3)		15.5 (10.25–18.75)	
	CT + TT	0.7 (0.5–1)	0.801 ^b^	262 (218.5–294.5)	0.060 ^b^	15 (12–17.25)	0.831 ^b^	0.18 (0.01–0.4)	0.500 ^b^	281 (225–376)	0.191 ^b^	15 (11–18)	0.550 ^b^
*CAT* rs1001179	CC	0.7 (0.4–1)	0.474 ^a^	240 (211–282)	0.248 ^a^	15 (13–17)	0.999 ^a^	0.2 (0.02–0.4)	0.441 ^a^	285.5 (222.3–373.3)	0.796 ^a^	14 (11.5–17.5)	0.757 ^a^
	CT	0.75 (0.6–1)		243.5 (221–276.8)		15 (12–17.25)		0.13 (0.01–0.2)		265 (231–433)		14.5 (10.75–19)	
	TT	0.5 (0.16–0.95)		301 (242–400.5)		15 (12–17.5)		0.4 (0.01–0.55)		243 (191–457)		15 (7–19.5)	
	CT + TT	0.7 (0.6–1)	0.698 ^b^	265 (222–296)	0.312 ^b^	15 (12–17)	1.0 ^b^	0.15 (0.01–0.35)	0.256 ^b^	264 (231–428)	0.701 ^b^	15 (10–19)	0.822 ^b^
*GPX1* rs1050450	CC	0.8 (0.4–1)	0.147 ^a^	240 (219–277)	0.874 ^a^	15 (13–17)	0.215 ^a^	0.2 (0.01–0.4)	0.698 ^a^	288.5 (231–394.8)	0.875 ^a^	14.5 (11–18)	0.583 ^a^
	CT	0.7 (0.43–0.9)		252 (217–293.5)		15 (12.5–17.75)		0.15 (0.01–0.3)		262 (223–376)		15 (11–19)	
	TT	1 (0.7–1)		246.5 (209–411.3)		13 (12–15)		0.2 (0.08–0.45)		291 (244.3–389.5)		14 (9.5–16)	
	CT + TT	0.7 (0.5–0.9)	0.655 ^b^	246.5 (217–298.5)	0.615 ^b^	15 (12–17)	0.583 ^b^	0.15 (0.01–0.4)	0.793 ^b^	267 (226–376)	0.853 ^b^	14.5 (10.75–18)	0.814 ^b^
*IL1B* rs1143623	GG	0.7 (0.5–1)	0.965 ^a^	252.5 (226.3–280.8)	0.536 ^a^	15 (13–17)	0.533 ^a^	0.2 (0.05–0.4)	0.541 ^a^	294 (231–393.8)	0.951 ^a^	14 (10.5–17.5)	0.686 ^a^
	GC	0.7 (0.4–1)		240 (208–286.5)		15 (12–17)		0.15 (0.01–0.3)		281 (222.8–389.5)		15 (11–18)	
	CC	0.9 (0.6–1)		270 (210.3–378.3)		14.5 (11.75–15.5)		0.06 (0.01–0.45)		259 (210.5–437.5)		15.5 (12.25–17.5)	
	GC + CC	0.7 (0.4–1)	0.627 ^b^	240 (210–298)	0.429 ^b^	15 (12–17)	0.357 ^b^	0.15 (0.01–0.35)	0.366 ^b^	281 (223.5–385)	0.949 ^b^	15 (11–18)	0.388 ^b^
*IL1B* rs16944	TT	0.85 (0.48–1)	0.637 ^a^	235 (214–321.3)	0.551 ^a^	14.5 (12–16.75)	0.384 ^a^	0.2 (0.01–0.4)	0.764 ^a^	273 (239.8–308.8)	0.755 ^a^	15 (11.5–18.5)	0.722 ^a^
	TC	0.7 (0.4–1)		240 (209.8–284.3)		15 (12–17)		0.15 (0.01–0.3)		286 (226.5–398)		14 (12–18)	
	CC	0.85 (0.6–1)		264 (224.8–285)		15 (13.75–18)		0.2 (0.02–0.4)		279.5 (227–393.8)		14.5 (10–17.75)	
	TC + CC	0.7 (0.5–1)	0.673 ^b^	246.5 (218.8–283)	0.965 ^b^	15 (13–17)	0.437 ^b^	0.2 (0.02–0.4)	0.909 ^b^	286 (228–395.5)	0.706 ^b^	14 (11–18)	0.719 ^b^
*IL1B* rs1071676	GG	0.8 (0.6–1)	0.070 ^a^	241 (218.3–286.5)	0.923 ^a^	15 (12–17)	0.754 ^a^	0.15 (0.01–0.38)	0.468 ^a^	281 (230.3–399.3)	0.513 ^a^	15 (11–17)	0.535 ^a^
	GC	0.7 (0.475–1)		257 (217–291)		15 (12.75–17.25)		0.2 (0.1–0.3)		300 (224–406)		14.5 (11.25–18)	
	CC	0.4 (0.18–0.55)		244.5 (205.3–278.5)		15 (13.5–17.25)		0.3 (0.01–0.7)		247.5 (231.3–310.8)		13 (8.5–16.5)	
	GC + CC	0.65 (0.4–0.93)	0.177 ^b^	257 (217–286)	0.920 ^b^	15 (13–17.25)	0.457 ^b^	0.2 (0.04–0.4)	0.261 ^b^	286 (228.5–379.5)	0.737 ^b^	14 (10.5–18)	0.675 ^b^
*MIR146A* rs2910164	GG	0.75 (0.53–1)	0.852 ^a^	247 (220.8–288.8)	0.627 ^a^	15 (13–17)	0.410 ^a^	0.15 (0.02–0.33)	0.377 ^a^	285 (231.5–389)	0.494 ^a^	15 (11–18)	0.745 ^a^
	GC	0.6 (0.43–1)		244 (211.5–286.8)		14 (12–16.5)		0.2 (0.06–0.4)		266 (225.3–375.8)		14 (11.25–17)	
	CC	0.7 (0.5–0.85)		224 (206–224)		15 (14.5–16.0)		0.05 (0.01–0.3)		380.5 (252.3–556.8)		14.5 (9–20)	
	GC + CC	0.6 (0.4–1)	0.580 ^b^	240 (211–279.5)	0.539 ^b^	14 (12–17)	0.236 ^b^	0.2 (0.01–0.4)	0.588 ^b^	274 (225.3–391.8)	0.869 ^b^	14 (11–17)	0.448 ^b^

^a^ additive model; ^b^ dominant model. BCVA—best corrected visual acuity; IOP—intraocular pressure; OCT-CRT—optical coherence tomography central retinal thickness; PVR—proliferative vitreoretinopathy; RRD—rhegmatogenous retinal detachment; SNP—single nucleotide polymorphism.

## Data Availability

Data available upon request.

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
