# Peer review of "Inflammation and Oxidative Stress Gene Variability in Retinal Detachment Patients with and without Proliferative Vitreoretinopathy"

_genes, 2023, doi:10.3390/genes14040804_

Round 1
Author Response
Reviewer 1
This study investigated the genetic association between single nucleotide polymorphisms (SNPs) involved in inflammation and oxidative stress pathways with proliferative vitreoretinopathy (PVR) in patients who underwent 3-port pars plana vitrectomy. The study analyzed 7 selected SNPs in 5 genes associated with response to oxidative stress or inflammation. The results showed significant differences in SOD2 and IL1B genotype frequencies between patients with and without PVR, with carriers of the polymorphic SOD2 rs4880 alleles having a higher risk of developing higher-grade PVR. The study suggests more extensive multicentric and population-based studies to confirm the findings further.
Overall, the manuscript is written correctly; however, some points need to be addressed, specifically:
-Abstract contains too much numerical data that is not essential to understand the concept and could be removed. Moreover, the abstract is too long and should be reduced to approximately 250 words. Finally, the language could be concise and easier to understand without sacrificing important information.
Response: We have now reduced the abstract to less than 250 words and tried to make it concise and easier to understand.
-Start the abstract with a clear and concise sentence like: "This study investigated the association between certain genetic variations and the risk of developing a proliferative vitreoretinopathy (PVR) after surgery."
Response: We have accepted the proposal and have now included it in the abstract.
-Explain BCVA and RRD abbreviations in the abstract
Response: The abbreviations in the abstract are now corrected and properly explained.
-Emphasize the key findings and their implications, such as "Our study suggests that certain genetic variations may play a role in the development of PVR after surgery. These findings may have important implications for identifying patients at higher risk and developing new treatments."
Response: We thank the reviewer for the valuable comments and suggestions. The sentence above certainly makes more sense and is now included in the abstract.
Moreover, in the introduction, there could be several improvements incorporated.
-The introduction is lengthy and can be shortened by focusing on the main points of the study.
Response: We have improved the Introduction. It is now more concise and focused on the main points of the study.
-There is no clear research question or hypothesis. Please add it.
Response: We have now added the research hypothesis.
Furthermore, I have a couple of suggestions (not required revisions) for the authors for consideration in this and future studies:
-The study would benefit from having a control group of healthy individuals without RD, providing a baseline comparison for the genetic associations investigated.
Response: A control group has now been added to the Results section and further elaborated in the text.
-More comprehensive genetic analysis: The study only investigated a small number of genetic variants in 5 genes involved in inflammation and oxidative stress. A more comprehensive analysis of genetic variants across a broader range of genes and pathways may provide a more complete understanding of the genetic basis of PVR.
Response: Although we agree with the reviewer that a more comprehensive genetic analysis may provide broader understanding of the genetic basis of PVR, such studies on many samples are very laborious and costly. At this stage, we have limited the analysis to the studied gene variants, both representing the inflammatory and oxidative stress pathways, which appear to be very important in PVR development.
- While the study conducted univariate analysis for each SNP, multivariate analysis could provide a better understanding of how different factors (age, sex, duration of symptoms, etc.) affect the genetic associations with PVR.
Response: The reviewer raises a good point in genomics research. We have done adjustments for age and gender in the current study, while in the revised Discussion we state the limitations of our findings.
- Investigating the functional significance of the identified genetic variants could help to understand the mechanisms by which they contribute to the development of PVR.
Response: This is another excellent idea, which needs to be tested in a separate study where one could possibly modify the inflammatory and oxidative milieu in the eye and see how functional outcome has been affected.
We greatly appreciate and are grateful for useful suggestions, which we have mostly been take into account or addressed accordingly, while other are planned in our further research.

Reviewer 2 Report
This study investigated the genetic association between single nucleotide polymorphisms (SNPs) involved in inflammation and oxidative stress pathways with proliferative vitreoretinopathy (PVR) in patients who underwent 3-port pars plana vitrectomy. The study analyzed 7 selected SNPs in 5 genes associated with response to oxidative stress or inflammation. The results showed significant differences in SOD2 and IL1B genotype frequencies between patients with and without PVR, with carriers of the polymorphic SOD2 rs4880 alleles having a higher risk of developing higher-grade PVR. The study suggests more extensive multicentric and population-based studies to confirm the findings further.
Overall, the manuscript is written correctly; however, some points need to be addressed, specifically:
-Abstract contains too much numerical data that is not essential to understand the concept and could be removed. Moreover, the abstract is too long and should be reduced to approximately 250 words. Finally, the language could be concise and easier to understand without sacrificing important information.
-Start the abstract with a clear and concise sentence like: "This study investigated the association between certain genetic variations and the risk of developing a proliferative vitreoretinopathy (PVR) after surgery."
-Explain BCVA and RRD abbreviations in the abstract
-Emphasize the key findings and their implications, such as "Our study suggests that certain genetic variations may play a role in the development of PVR after surgery. These findings may have important implications for identifying patients at higher risk and developing new treatments."
Moreover, in the introduction, there could be several improvements incorporated.
-The introduction is lengthy and can be shortened by focusing on the main points of the study.
-There is no clear research question or hypothesis. Please add it.
Furthermore, I have a couple of suggestions (not required revisions) for the authors for consideration in this and future studies:
-The study would benefit from having a control group of healthy individuals without RD, providing a baseline comparison for the genetic associations investigated.
-More comprehensive genetic analysis: The study only investigated a small number of genetic variants in 5 genes involved in inflammation and oxidative stress. A more comprehensive analysis of genetic variants across a broader range of genes and pathways may provide a more complete understanding of the genetic basis of PVR.
- While the study conducted univariate analysis for each SNP, multivariate analysis could provide a better understanding of how different factors (age, sex, duration of symptoms, etc.) affect the genetic associations with PVR.
- Investigating the functional significance of the identified genetic variants could help to understand the mechanisms by which they contribute to the development of PVR.
Author Response
Responses to Reviewer 2 comments
Dear authors, this is an interesting study about different SNPs studied for their association with PVR grade C1 or higher. It is well written and provide original results in this research field. On the other hand, I found some aspects that might be improved, especially regarding the provided results and the criteria used for the inclusion of SNPs in the study.
Please, find my comments below, I hope these may help to improve your manuscript:
- RRD is not defined in the abstract.
Response: The definitions and abbreviations have been corrected accordingly.
- In the sentence (abstract): “Furthermore, we evaluated the possible association of SNPs with postoperative clinical using…” Please, include the word “parameters” between “clinical” and “using”.
Response: The sentence has now been corrected and the word “parameters” is included accordingly.
- It might be provided some information about how the included SNPs were chosen. All of them have high variant allele frequencies, was this an inclusion criteria? Or was this a discretional decision? Why these SNPs and not others? E.g. you talk about the VEGF but you don´t include any SNP of the VEGFA gene (or KDR/FLT1 encoding its receptors).
Response: Five candidate genes were selected based on their direct involvement in response to oxidative stress and inflammation pathways. Only common experimentally confirmed- or in silico predicted- functional polymorphisms with minor allele frequency above 5% in the European population were included in the study: non-synonymous SNPs located within the coding region or SNPs in the vicinity of selected genes (5’ and 3’ untranslated region) that could alter expression levels. Seven SNPs fulfilling these criteria were selected for genotyping: rs4880 (SOD2); rs1001179 (CAT); rs1050450 (GPX1); rs1143623, rs16944, rs1071676 (IL1B); and rs2910164 (MIR146A).
- It should be included a “study limitations” section. In this section it might be commented the SNPs inclusion criteria procedures if it did not follow an unbiased criterion.
Response: The “study limitations” sections is now included in the manuscript.
- Genes must be written in italics. Please, review and amend this throughout the manuscript. e.g.: TGFB1 and IL10 in the introduction section.
Response: All genes throughout the manuscript are now written in italics.
- Please, review the bibliography. I cannot find the reference “18” in the introduction section.
Response: We have reviewed the bibliography. The reference 18 is cited in the second paragraph at the Introduction, between the references 14-19.
- Table 2 might include data about variant/minor allele frequencies of included SNPs from other sources (e.g. European population in the 1000 genomes project), and a comparison with your population to discard possible recruitment bias in the study.
Response: VAF (1000G): variant allele frequency reported in European population in the 1000Genomes project in dbSNP. *comparison of variant allele frequency with 1000Genomes project using Chi square test are now reported and included in the Table2.
- Regarding Table 3. You comment: “Carriers of the two polymorphic SOD2 rs4880 alleles (TT genotype) were more likely to have higher grade PVR (OR=2.31, 95% CI=1.01-5.27, P=0.046)”. Based on table 3, I understand that you compared the TT vs CC genotypes. In my opinion you should say: “Carriers of the two polymorphic SOD2 rs4880 alleles (TT genotype) were more likely to have higher grade PVR (OR=2.31, 95% CI=1.01-5.27, P=0.046) compared to CC genotype”, and include results for further genetic models (recessive, dominant, co-dominant, over-dominant), especially the comparison of TT vs CC/CT genotypes for SOD2 rs4880.
Response: We thank the reviewer for the valuable suggestions. The suggested sentence is now included in the text.
- I cannot find Table 4 (cited in the results section).
Reference: Table 4 was too wide to fit in a readable format in the text, therefore, we had to place it at the end of the manuscript after the references.
- In my opinion, it would be interesting to enhance the results with an Allele association study, not just a genotype association study. I understand this may be much time consuming thus might be commented as a future study to be done.
Response: Indeed, this is our goal for a future research
- Dear authors, you compare the “study group” (PVR grade C1) with a control group of patients without grade C1. It would be very interesting to compare both groups with another control group of random population. This may support your results if you find differences between your population and this “control” group. Also, it may occur that some of the included SNPs are associated with PPV and RRD, and not with PVR. I understand this is not the main endpoint of your study and much time consuming to be done now that you have this article written. Anyway, it may be commented in the discussion section as a limitation or a future perspective of your research.
Response: We thank the reviewer for this useful suggestions. The comment is now added as a limitation of the study.
- Also, related to my previous comment. It would be very interesting to discuss the pharmacogenetic or pharmacogenomics consequences/aspects of your study. I understood that you found new genetic biomarkers of PVR. But which may be the consequences of this results on the clinical practice? Might be these SNPs used in the future to take a decision on the clinical management of patients with PPV and RRD? In my opinion, discussing these points will enhance the interest of readers and researchers in this field.
Response: Thank you for very interesting point. We have now added a discussion on the possible implications of the results of the study in future research and clinical practice.
- Please change: “… an association of the SOD2 rs4880 (TT) and IL1B rs1071676 (GC) genes” to “… an association of the SOD2 rs4880 (TT) and IL1B rs1071676 (GC) SNPs (or variants)”
Response: Corrected.
- Regarding the conclusions: You wrote, “Our results demonstrate an association of the SOD2 rs4880 (TT) and IL1B rs1071676 (GC) genes with the development of PVR C1 or higher, which can therefore be considered as unfavourable prognostic factors.” In my opinion you should say: “Our results demonstrate an association with the development of PVR C1 or higher of the SOD2 rs4880 TT genotype compared to CC genotype, and IL1B rs1071676 GC compared to GG genotype.”
Response: Corrected.
- You also say: “This can therefore be considered as unfavourable prognostic factors.” In my opinion, this sentence referred to the IL1B rs1071676 must be further discussed since, as you comment in the results section, “no differences could be observed in the univariable analysis or in the dominant model (all P>0.050)”, and the Hardy-Weinberg equilibrium analysis for this SNP showed a p-value= 0.062 (close to significant). As commented above, including results for at least the recessive and dominant model for each SNP may help in the understanding of your results.
Response: In this case more patients and controls are needed to clarify this issue.
